# Non-Alcoholic Fatty Liver Disease and Metabolic Syndrome in Women: Effects of Lifestyle Modifications

**DOI:** 10.3390/jcm11102759

**Published:** 2022-05-13

**Authors:** Maria Teresa Guagnano, Damiano D'Ardes, Rossi Ilaria, Francesca Santilli, Cosima Schiavone, Marco Bucci, Francesco Cipollone

**Affiliations:** 1“Clinica Medica” Institute, Department of Medicine and Aging Science, “G. D’Annunzio” University of Chieti-Pescara, Vestini Road, 66100 Chieti, Italy; guagnano@unich.it (M.T.G.); rossilaria91@gmail.com (R.I.); fra.santilli22@gmail.com (F.S.); mbucci@unich.it (M.B.); francesco.cipollone@unich.it (F.C.); 2Unit of Ultrasound, Department of Medicine and Aging Science, “G. D’Annunzio” University of Chieti-Pescara, Vestini Road, 66100 Chieti, Italy; cosima.schiavone@unich.it

**Keywords:** NAFLD, metabolic syndrome, diet, obesity, lifestyle modifications, ultrasound, elastosonography

## Abstract

Non-alcoholic fatty liver disease (NAFLD) is the most widespread liver disease, characterized by fatty acids liver accumulation and subsequent fibrosis. NAFLD prevalence ranges from 80% to 90% in obese subjects and is estimated to be around 50% in patients with metabolic syndrome. In this clinical scenario, diet and lifestyle modifications can play an important role. There are several imaging techniques that can accurately diagnose fatty liver. Recently, ultrasound has acquired a leading role in the diagnosis and follow-up of fatty liver disease. Furthermore, elastosonography represents a valid alternative to liver biopsy. Shear wave elastosonography evaluates the elastic and mechanical properties of liver tissue. The aim is to evaluate the effects of lifestyle and nutritional interventions and a loss of body weight during hepatic steatosis through ultrasonographic and elastosonographic techniques. Thirty-two female subjects with metabolic syndrome were subjected to clinical, anthropometric, and laboratory assessments, as well as abdominal ultrasonographic/elastosonographic measurements taken from enrollment time (T0) and after 3 months (T1) of lifestyle modifications. After 3 months of lifestyle changes, significant weight loss was observed, with a marked improvement in all adiposity indices. The laboratory parameters at T1 showed significant decreases in total and LDL cholesterol, triglycerides, basal blood glucose, 120 min glycaemia, basal insulin and HOMA Index (*p* < 0.001). A similar improvement was observed at T1 for steatosis degree (*p* < 0.01) and elastosonographic measurements (Kpa *p* < 0.001). The linear regression analysis of the baseline conditions documented that the size of the liver positively correlated with body weight, BMI, neck and waist circumferences, waist to height ratio (WhtR), insulin and HOMA Index, fat mass and visceral fat, and steatosis grade. After 3 months, the liver size showed improvement with positive correlations to all previous variables. Hepatic stiffness (Kpa) positively correlated with neck circumference, visceral fat, and ALT, with basal insulin, gamma-GT, and AST, and with waist circumference, WhtR, and fat mass. The degree of steatosis was positively correlated with more variables and with greater statistical significance at T1 with respect to T0. Particularly, the positive correlations between the degree of steatosis and neck circumference (*p* < 0.001), HOMA Index, and triglycerides (*p* < 0.001) appeared to be very significant. NAFLD management in women with metabolic syndrome should be focused on lifestyle modifications. Moreover, liver involvement and improvement at follow-up could be evaluated in a non-invasive manner through ultrasonographic and elastosonographic techniques.

## 1. Introduction

Non-alcoholic fatty liver disease (NAFLD) is the most widespread liver disease worldwide, characterized by fatty acids liver accumulation and subsequent fibrosis. Normally, fat represents less than 5% of the liver weight; when this percentage is higher, steatosis occurs [1,2,3]. The NAFLD prevalence is around 25% in Italy (70–80% in obese subjects and type 2 diabetes) and 30% in North America. The highest prevalence is recorded in the Middle East (32%), South America (31%), and Asia (27%), while the lowest is reported in Africa (14%) [4,5]. Moreover, NAFLD is related to a broad range of liver parenchyma damage. Simple steatosis has a low risk of progression into cirrhosis, while a significant percentage of subjects with NAFLD (10–15%) have histological features of necroinflammation and balloon-like degeneration characterizing the most severe form of liver disease: Non-Alcoholic Steatohepatitis (NASH), with a possible evolution into fibrosis, cirrhosis, and hepatocellular carcinoma (HCC) [6,7,8,9]. The majority of NAFLD patients have metabolic comorbidities, such as diabetes, obesity, and dyslipidemia [10]. NAFLD prevalence ranges from 50% to 75% in type 2 diabetes mellitus [11,12], from 80% to 90% in obese subjects [13,14], and is estimated to be around 50% in patients with metabolic syndrome [15], while the prevalence of metabolic syndrome in NAFLD and NASH patients is reported at 43% and 71%, respectively [16]. The so-called ‘metabolic syndrome’ (MS), defined on the basis of the combination of central obesity, impaired glucose metabolism, atherogenic dyslipidemia, and arterial hypertension, is present in the large majority of subjects affected by visceral obesity, and it is the major risk factor predisposing the NAFLD and NASH. Indeed, these metabolic perturbations contribute to the molecular pathogenesis of NAFLD [3,17,18,19,20,21,22]. Increased insulin resistance promotes the early stages of hepatic steatosis, especially through the increased mobilization of fatty acids from visceral adipose tissue to the liver and subsequent deposition of triglycerides in hepatocyte cytoplasm [23]. There are several imaging techniques that can accurately diagnose fatty liver, but there is currently no reliable means for detecting NASH or early cirrhosis [24,25,26]. Recently, ultrasound has a leading role in the diagnosis and follow-up of fatty liver disease [27]. Liver steatosis is characterized by the accumulation of triglycerides within the intrahepatocytic micro- and macrovesicles. The lipid vacuoles form several interfaces reflecting ultrasounds, thus generating hyperechoic liver or bright liver. In steatosis, the liver has a smooth, regular surface with rounded edges and usually has an increased volume. The most commonly used measurement is the longitudinal diameter of the right lobe (12–13 cm) [28]. The fibrosis is itself a cause of hyper-echogenicity such as steatosis, with which it often coexists, the “fatty fibrotic liver”, to indicate that the two forms are not always ultrasonographically differentiable [27]. Ultrasound, as well as CT and MRI, does not allow for discriminating patients with steatosis from those with developmental steatohepatitis, except in the already evolved forms [28]. Ultrasound does not provide information regarding the necro-inflammatory activity and mechanical properties of the liver tissue, that is, its rigidity. Liver fibrosis is indeed characterized by a greater “hardness” of the hepatic parenchyma, and the gold standard for diagnosis is liver biopsy [29,30]. Elastosonography represents a valid alternative to liver biopsy. Shear wave elastosonography evaluates the elastic and mechanical properties of liver tissue. This method exploits the fact that many pathologies cause a tissue stiffness change [28].

The aim of the study was to assess obese female subjects with metabolic syndrome for the presence and grade of NAFLD. In addition, the study evaluated the effect of body weight loss on the improvement of hepatic steatosis by ultrasonography and elastosonography with the shear wave technique after 3 months of lifestyle- nutritional intervention.

## 2. Materials and Methods

### 2.1. Study Design

Thirty-two white European female subjects who spontaneously attended the Obesity Centre of “G. D’Annunzio” University of Chieti between May and November 2020 to be subjected to a structured nutritional assessment were recruited. The inclusion criteria were female sex, age ≥ 18 ≤ 70 years, and the presence of Metabolic Syndrome (according to the International Federation of Diabetes-IDF) [20]. The following were exclusion criteria: bariatric surgery, neurological and/or psychiatric pathologies, oncological therapy, and secondary liver disease. The presence of NAFLD was not one of the inclusion criteria. Each participant was subjected to clinical, anthropometric, and laboratory assessment, as well as abdominal ultrasonography measurements during the same morning, at enrollment time (T0), and after 3 months (T1) of lifestyle modifications.

During the study period, four subjects dropped out, and two subjects were hospitalized due to other acute diseases. The final dataset included a total of 26 subjects.

Ongoing therapies have not been changed during the 3 months of observation. None of the subjects took antidiabetic and/or lipid-lowering drugs or were on a low-calorie diet. Furthermore, none of the subjects recruited at the baseline were diabetic.

### 2.2. Clinical and Anthropometric Measurements

Patients underwent clinical-anthropometric evaluation and fasting in the morning; weight was measured in kilograms and height in centimeters, BMI in kg/m^2^, neck, waist, and hip circumference in centimeters, and systo—diastolic blood pressure in mmHg, according to the World Health Organization guidelines [31,32,33]. The subjects were weighed without shoes and in light clothing, with an approximation of 0.1 kg; the height was measured with an approximation of 0.5 cm. The neck circumference was measured with an extendable centimeter tape, passing posteriorly, from the midpoint of the cervical tract and anteriorly just below the laryngeal prominence. The subject’s head was held erect, with the eyes facing forward and the neck in a horizontal plane at the level of the most prominent portion, i.e., the thyroid cartilage. The waist circumference was measured with an extendable centimeter tape at the intermediate point of the line that joins the xiphoid to the iliac crest, with the subject standing and breathing normally. The hip circumference was measured with an extendable centimeter tape at the level of the greater trochanter. The ratio of the waist to hip circumference (WHR) and the ratio of the waist circumference to height (WhtR) were also calculated. Blood pressure was assessed after 15 min of rest in a seated position on the upper left hand, and the systolic (SBP) and diastolic (DBP) blood pressure were also collected [34]. Each participant in the study performed a body impedance test with a Body Composition Analyzer (BIA)—SC-330-(Tanita)—Milan—ITA [35] to evaluate the fat mass, lean mass, basal metabolism, and visceral fat, expressed in levels (range 1–59; values > 13 considered as the threshold for a major risk) [35].

### 2.3. Laboratory Data

The following laboratory tests were performed: Fasting glucose, the Oral Glucose Tolerance Test, (OGTT-75 g) [36], total cholesterol, HDL-cholesterol, triglycerides, LDL-cholesterol (according to Friedwald’s formula: LDL cholesterol = Total Cholesterol − (HDL cholesterol + Triglycerides/5) [37], insulin levels, HOMA Index according to the formula: blood glucose (mg/100 mL) × insulinemia (mUI/L)/405 [38,39], uric acid, alanino-amino transferase (ALT), aspartate amino transferase (AST), and gamma glutamil transferase (yGT).

### 2.4. Lifestyle Modifications—Dietary Regimens Protocol

A personalized hypocaloric Mediterranean diet (1400–1800 calories), adequate water intake (2 L/day still water), and moderate daily aerobic physical activity (30 consecutive min/day for at least five times/week) were prescribed to all subjects [40,41].

The dietary regimen was established according to current guidelines for a balanced composition of macronutrients (56% carbohydrates, 17% protein, and 27% fat), the daily intake of cholesterol was (<300 mg/die), daily intake of saturated fatty acids (<10% of total energy intake), daily intake of oligosaccharides (<15% of total energy intake), and the daily intake of dietary fiber (25–30 g/die). Moreover, the total protein intake was 50% from animal and 50% from vegetable proteins [42,43,44].

### 2.5. Ultrasound and Elastosonography Evaluation

All of the patients underwent abdominal ultrasound with a Philips 7G-ultrasound system integrated by the PQ and Q elastosonographic technique, using a 3.5–5 MHz C5-1 convex way probe [27]. The ultrasound evaluation and elastosonography were performed after the patient fasted for 8 h, in the supine position with the arms behind the head and the operator positioned to the right of the table. The probe was positioned perpendicular to the skin, and the B-mode image was obtained through the right trans-costal acoustic window [27]. Data on the size of the hepatic right lobe and on the degree of steatosis were acquired. To evaluate the steatosis, hepatic and renal echogenicity were compared. The steatosis was evaluated according to the following scale: 1-absent steatosis, 2-mild steatosis, 3-moderate steatosis, and 4-severe steatosis.

The elastosonographic data were shown as a mean of ten technically correct measurements through intercostal scans, at about 2.5 cm from the liver capsule, trying to avoid sampling of vascular or biliary structures. Measurements had a variability below 30%, and the median of the measurements was calculated and classified according to the Metavir scale (used to classify fibrosis with liver biopsy histologically), based on the kPa range, measured with shear wave EPQ [45].

After 3 months of lifestyle intervention, anthropometric parameters, laboratory tests, upper abdomen ultrasound, and elastosonography were remeasured.

### 2.6. Statistical Analysis

The descriptive data for the main variables are reported as mean ± standard deviation.

The paired-t-test was used to compare the variables between the T0 and T1. The Pearson correlation coefficients were also calculated to assess the relationships between the variables. *p* < 0.05 was considered the significance level. All of the statistical analyses were performed using the R software environment for statistical computing and graphics, version 3.5.2 (R Foundation for Statistical Computing, Vienna, Austria).

The study complied with the principles established by the Declaration of Helsinki, and written informed consent was obtained from each subject. The study was approved by the Ethics Committee of the Provinces of Chieti and Pescara and of the “G. d’Annunzio” University of Chieti-Pescara (Ethics Committee Project n.7—14 May 2020).

## 3. Results

The clinical-anthropometric characteristics of the patients at baseline (T0) and after 3 months (T1) are shown in Table 1. All subjects reported having observed lifestyle modifications. The average age of the subjects was 49.00 + 13.43 years, with an average duration of obesity of 12.19 + 12.02 years. After 3 months of lifestyle changes, significant weight loss has been observed, with a marked improvement in all adiposity indices (weight; BMI; waist, hip, and neck circumferences; fat mass; visceral levels; WhtR) (*p* < 0.001). At T1, there was also a decrease in blood pressure (SBP and DBP) (*p* < 0.001). The laboratory parameters, after 3 months of dietary-behavioral treatment, showed significant decreases for total and LDL cholesterol (*p* < 0.001), triglycerides (*p* < 0.001), basal blood glucose and at 120 min (after oral glucose tolerance test) (*p* < 0.001), basal insulinemia and HOMA Index (*p* < 0.001), and gamma-glutamyl transferases (*p* < 0.01) (Table 2). A similar improvement has been observed at T1 for steatosis degree (*p* < 0.01) and elastosonographic measurements (kilopascal *p* < 0.001) (Table 3). After 3 months, 21 subjects still had NAFLD and 12 subjects (46.15%) no longer presented Metabolic Syndrome. In particular, after 3 months of lifestyle modifications: 4 obese women became overweight, 17 women had BP levels <140/90 mmHg, 12 subjects showed normal glycemic levels, and 23 subjects (88.46%) had total cholesterol levels <200 mg/dL. The NAFLD was present in 88.46% before and 80.76% after the intervention. None of the subjects had cirrhosis before and after 3 months. In four women (15.38%), the degree of fibrosis remained unchanged, and in 22 subjects (84.61%), there was a decrease in the degree of fibrosis after the intervention. The linear regression analysis at baseline conditions documented that the size of the liver correlated positively with body weight (*p* = 0.03); BMI (*p* = 0.02); neck and waist circumferences (*p* = 0.001 and *p* = 0.03); WhtR (*p* = 0.03); insulin (*p* = 0.03) and HOMA Index (*p* = 0.04); fat mass (*p* = 0.02) and visceral fat (*p* = 0.03); steatosis grade (*p* = 0.008). After 3 months, the liver size showed improvement and positive correlations with all previous variables. In addition, liver size presented positive correlations with hip circumference and with SBP (*p* = 0.01) and negative correlations with HDL-cholesterol (*p* = 0.04) (Table 4). Hepatic stiffness positively correlated with neck circumference, visceral fat and ALT (*p* = 0.01), with basal insulin, gamma-GT and AST (*p* = 0.03) and with waist circumference, WhtR, fat mass (*p* = 0.04). After 3 months of treatment, positive correlations persisted between hepatic stiffness and AST (*p* = 0.01), basal insulin (*p* = 0.02), and visceral fat (*p* = 0.04). Hepatic stiffness positively correlated with HOMA Index at T1(*p* = 0.03) (Table 4). Finally, the degree of steatosis had statistically significant direct correlations at T0 with most of the anthropometric-laboratory variables, and at T1, the degree of steatosis was positively correlated with more variables and with a greater statistical significance, as shown in Table 4. In particular, the positive correlations between the degree of steatosis and neck circumference (r = 0.7014; *p* < 0.001), HOMA Index, and triglycerides (*p* < 0.001) was very significant (Table 4).

## 4. Discussion

The present study focuses on nonalcoholic fatty liver disease and metabolic syndrome. Both are determined by the expansion of visceral adipose tissue. Abdominal obesity (measured as waist circumference—WC ≥ 88 cm in women and ≥102 cm in men) is the primary factor in metabolic syndrome independent of body mass index [46]. The results highlighted a marked improvement after 3 months of correct dietary habits in all metabolic and anthropometric parameters, including ultrasound and elastosonographic ones, proving that a radical change in lifestyle mainly affects weight loss and consequently metabolic parameters such as liver steatosis. It is necessary to observe more significant results at a time of >3 months. Our subjects had an average BMI of 39.17, a borderline value between the second- and third-degree of obesity.

Several researchers have clearly shown that the prevalence of metabolic syndrome is increasing worldwide as obesity rates continue to grow [20,47,48,49]. Moreover, visceral adiposity and hepatic steatosis (and NAFLD in general) have been shown to be key factors in metabolic syndrome [50,51,52]. NAFLD is the most common cause of hepatic steatosis by far and is known to be associated with the characteristics of metabolic syndrome and cardiovascular disease, but it has yet to be determined whether it is a cause or an effect [53,54]. It was recently observed by liver biopsy that steatohepatitis represents the sole feature of liver damage in type 2 diabetes. This observation confirms the hypothesis that T2DM and insulin resistance status increase the risk of advanced fibrosis, with a consequential worsening of hepatic outcomes [55]. Moreover, insulin resistance is the strongest pathophysiological link between NAFLD and Metabolic Syndrome. Recent studies have shown that the reduction in insulin resistance through the pharmacological eradication of HCV by direct-acting antivirals leads to both a reduction in the onset of type 2 diabetes and clinical expressions of atherosclerosis [56,57,58]. NAFLD and insulin resistance are bidirectionally correlated. One very recent review explains in an updated and complete way the pathophysiological mechanisms that support this relationship [59].

Genetic predisposition and epigenetics cannot fully explain the disease onset or the rise in NAFLD prevalence observed in Western countries over the last decades. Environmental factors, such as dietary habits and physical activity, and also gender, have been shown to play a significant pathophysiological role in NAFLD [10,29,60]. There is evidence that the expanded visceral adipose depot is a source of cytokines and adipokines deeply involved in the metabolic, vascular, and immunological homeostasis by paracrine and endocrine mechanisms [61,62,63,64,65,66].

In our subjects, the adherence to a traditional Mediterranean diet, characterized by the consumption of antioxidant-rich foods, can be considered a good approach for the treatment of NAFLD. The worldwide spread of NAFLD diagnosis is clearly linked to changes in dietary profiles and increased sedentary lifestyle, not only in Western countries but also in the urban area of developing countries [67]. International recommendations indicate that the first therapeutic step for the treatment of NAFLD is to reduce the intake of total fat, saturated fatty acids, trans-fatty acids, and fructose, along with undergoing physical activity [67]. A recent study by Baratta et al. showed that the Mediterranean diet reduces the risk of NAFLD [40].

Other studies have discussed the relationship between food intake and fatty liver or its related conditions. In a cross-sectional study, Williams et al. reported that a balanced diet accompanied by the frequent consumption of raw vegetables, salad, fruit, fish, pasta, rice, and low consumption of fried foods, such as sausages, fried fish, and potatoes is negatively related to abdominal obesity, glucose, plasma triglyceride, and positively related to HDL levels [3,68].

Recent studies have shown that the increased consumption of fruits and vegetables reduces the risk of heart attacks, ischemic heart diseases, hypertension, and type 2 diabetes and contributes to weight loss [69,70]. Other studies have shown that an increased fat intake and the Western diet are associated with insulin resistance and the progression of NAFLD [71,72,73].

The role of dietary composition in modifying the onset and severity of NAFLD has been shown in population-level studies, where NAFLD patients were commonly presenting with unhealthy eating habits (i.e., eating processed foods, frequently eating at restaurants), shallow levels of physical activity, and higher sedentary behavior [74]. Conversely, an active lifestyle and a higher consumption of fruits and vegetables are linked to a lower risk of NAFLD [75,76]. Moreover, lifestyle-induced weight loss is found to improve liver histology and function, as well as cardiometabolic profile, among NAFLD patients [77,78].

The relationship between neck circumference and metabolic syndrome has been demonstrated in our study, as other authors have also noted [79]. In our patients, neck circumference has been significantly associated with the occurrence of NAFLD compared with other anthropometric indices [75]. Neck circumference is more feasible, accessible, has fewer limitations, excellent repeatability, and minimal variance during the day [80]. Neck circumference is accepted as an alternative measurement to detect fat accumulation in the upper body, a finding which is considered to be indicative of a significant metabolic risk factor for type 2 diabetes mellitus and hyperlipidemia in adults [81,82,83].

Unfortunately, our study also has some limits, particularly regarding gender disparity and the limited number of patients. On the contrary, the gender disparity could represent an advantage because women are usually less represented in medical studies. Moreover, our study is probably the first to show how non-invasive ultrasonographic and elastosonographic techniques could help clinicians to measure the liver’s involvement in metabolic syndrome in women and monitor the follow-up and improvement caused by a therapeutic approach constituted by lifestyle modifications.

## 5. Conclusions

Our data show that NAFLD management in women with metabolic syndrome should be focused on lifestyle modifications. Moreover, liver involvement and improvement at follow-up could be evaluated in a non-invasive manner through ultrasonographic and elastosonographic techniques. Our data could also underline how critical it is to prevent NAFLD with more educational interventions to explain the importance of observing a healthy dietary regimen. However, the prevention of NAFLD should begin in subjects with overweight or with an initial metabolic syndrome. Regardless, further studies are needed to confirm our preliminary data and to better elucidate the complex interaction between NAFLD and metabolic syndrome in order to develop new therapeutic strategies.

## Figures and Tables

**Table 1 jcm-11-02759-t001:** Clinical characteristics before and after life style modifications (Mean ± Standard Deviation).

n. 26	BASAL	*p*-Value	III Month
Age (years)	49.00 ± 13.43	/	/
Duration of obesity (years)	12.19 ± 12.02	/	/
Weight (kg)	102.05 ± 17.87	0.001	94.853 ± 17.05
BMI (kg/m^2^)	39.17 ± 7.06	0.001	36.41 ± 6.80
Neck circumference (cm)	39.76 ± 2.77	0.001	38.57 ± 3.03
Waist circumference (cm)	121.50 ± 13.96	0.001	115.57 ± 14.20
Hip circumference (cm)	123.73 ± 11.38	0.001	119.23 ± 11.56
WHR	0.98 ± 0.09	NS	0.97 ± 0.09
WhtR	0.75 ± 0.09	0.001	0.71 ± 0.09
Fat Mass (kg)	48.46 ± 11.09	0.001	43.91 ± 11.41
Free Fat Mass (kg)	50.06 ± 6.36	NS	49.66 ± 6.70
Visceral Fat (levels)	13.53 ± 3.19	0.001	12.03 ± 3.32
SBP (mmHg)	149.42 ± 9.72	0.001	125.76 ± 12.05
DBP (mmHg)	88.26 ± 6.62	0.001	78.26 ± 8.93

BMI: Body mass index; Visceral Fat Levels range 1–59; SBP: Systolic Blood Pressure; DBP: Diastolic Blood Pressure; WHR: Waist to Hip Ratio; WhtR: Waist to height Ratio; NS: Not Significant.

**Table 2 jcm-11-02759-t002:** Laboratory parameters before and after lifestyle modifications (Mean ± Standard Deviation).

n. 26 Patients	BASAL	*p*-Value	III Month
Total Cholesterol (mg/dL)	234.38 ± 25.77	0.001	182.30 ± 28.59
HDL Cholesterol (mg/dL)	49.80 ± 12.89	NS	50.00 ± 12.24
Triglycerides (mg/dL)	183.46 ± 67.52	0.001	137.19 ± 42.81
LDL cholesterol (mg/dL)	147.88 ± 30.57	0.001	104.86 ± 29.77
FBG (mg/dL)	124.96 ± 14.00	0.001	102.30 ± 12.63
Blood Glucose 120′ (mg/dL)	163.11 ± 28.05	0.001	135.61 ± 21.60
Insulin (μU/mL)	21.65 ± 14.28	0.001	16.00 ± 9.73
HOMA Index	6.72 ± 4.50	0.001	4.12 ± 2.75
ALT (U/L)	21.76 ± 8.49	NS	20.46 ± 7.57
AST (U/L)	29.80 ± 15.38	NS	28.46 ± 15.26
γGT (U/L)	28.23 ± 15.58	0.01	23.65 ± 15.46
Uric Acid (mg/L)	5.50 ± 1.03	NS	5.43 ± 0.91

FBG: Fasting Blood Glucose; HOMA index: Homeostasis Model Assessement; ALT: Alanino-Amino Transferase; AST: Aspartate Amino Transferase; yGT: Gamma glutamil transferase; NS: Not Significant.

**Table 3 jcm-11-02759-t003:** Ultrasound and elastosonographic parameters before and after lifestyle modifications (Mean ± Standard Deviation).

n. 26 Patients	BASAL	*p*-Value	III Month
Liver size (cm)	15.26 ± 2.15	NS	14.60 ± 2.16
Grade of steatosis (1–4)	2.73 ± 0.86	0.01	2.38 ± 0.84
kPa (kiloPascal)	4.69 ± 1.04	0.001	3.89 ± 0.80

NS: Not Significant.

**Table 4 jcm-11-02759-t004:** Linear regressions between liver measurements and clinical-laboratory parameters at Basal and after 3 months of lifestyle modification (Pearson) (n. 26 patients).

Dependent Variable	Independent Variable	BASAL r *p*	III Month r *p*
Liver size	Weight	0.3643 0.03	0.6390 0.0001
	BMI	0.3981 0.02	0.5329 0.002
	Waist Circumference	0.3682 0.03	0.5698 0.001
	Hip Circumference	0.2126 NS	0.4400 0.01
	Neck Circumference	0.5598 0.001	0.7442 0.00006
	WhtR	0.3613 0.03	0.4461 0.01
	Fat Mass	0.3768 0.02	0.5892 0.0007
	Visceral Fat	0.3595 0.03	0.4081 0.01
	Insulin	0.3686 0.03	0.4389 0.01
	HOMA Index	0.3413 0.04	0.3520 0.03
	HDL	−0.1294 NS	−0.3469 0.04
	GammaGT	0.3133 0.05	0.3986 0.02
	Steatosis grade	0.4639 0.008	0.6248 0.0003
	SBP	0.1552 NS	0.4393 0.01
Hepatic stiffness	Waist Circumference	0.3431 0.04	0.1941 NS
	Neck Circumference	0.4055 0.01	0.2445 NS
	WhtR	0.3338 0.04	0.1636 NS
	Fat Mass	0.3317 0.04	0.2491 NS
	Visceral Fat	0.4203 0.01	0.3338 0.04
	Insulin	0.3550 0.03	0.3884 0.02
	HOMA Index	0.3050 NS	0.3638 0.03
	GammaGT	0.3572 0.03	0.2802 NS
	ALT	0.4196 0.01	0.1916 NS
	AST	0.3633 0.03	0.4261 0.01
Steatosis grade	Age	0.3898 0.02	0.1292 NS
	Weight	0.1237 NS	0.5121 0.003
	BMI	0.1086 NS	0.4367 0.01
	Waist Circumference	0.2240 NS	0.4599 0.009
	Neck Circumference	0.4239 0.01	0.7014 0.00003
	WhtR	0.3051 NS	0.3725 0.03
	Fat Mass	0.2202 NS	0.5772 0.001
	Visceral Fat	0.3453 0.04	0.5379 0.002
	Triglycerides	0.2630 NS	0.5882 0.0007
	FBG	0.3798 0.02	0.0153 NS
	Blood Glucose 120′	0.3466 0.04	0.4248 0.01
	Insulin	0.4201 0.01	0.6475 0.0001
	HOMA Index	0.4622 0.008	0.5833 0.0008
	ALT	0.3951 0.02	0.2318 NS
	GammaGT	0.5296 0.002	0.5663 0.001
	SBP	0.2308 NS	0.5025 0.004

BMI: Body mass index; SBP: Systolic Blood Pressure; yGT: Gamma glutaril transferase; AST: Aspartate Amino Transferase; ALT: Alanino-Amino Transferase; FBG: Fasting Blood Glucose; HOMA Index: Homeostasis Model Assessement Index; WhtR: Waist to height Ratio; NS: Not Significant.

## Data Availability

The data that support the findings of this study are available from the corresponding author upon reasonable request.

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
