# Peer review of "Non-Alcoholic Fatty Liver Disease and Metabolic Syndrome in Women: Effects of Lifestyle Modifications"

_jcm, 2022, doi:10.3390/jcm11102759_

Round 1

Reviewer 1 Report

I enjoyed reviewing this interesting manuscript. The paper is quite original and well written. However, I raise some issues that need to be addressed.

1- The population studied has an average BMI of 39.17, a borderline value between second- and third-degree obesity. Therefore, the results described by the authors on the favorable effect of a lifestyle modification on NAFLD, although very interesting, are not generalizable to a population with a lower BMI.

2- Neither the text nor the tables indicate how many subjects recruited at the baseline were diabetic. Please, add this information, and, if possible, HbA1c values both at the baseline and after the lifestyle intervention. Moreover, add a comment if appropriate.

3- About that, it was recently observed by liver biopsy that steatohepatitis represents the sole feature of liver damage in type 2 diabetes (PLoS One. 2017 Jun 1;12(6):e0178473. doi: 10.1371/journal.pone.0178473.). This observation confirms the hypothesis that T2D and IR status increase the risk of advanced fibrosis, with the consequent worsening of hepatic outcomes. This important issue should be commented on in the discussion, and the above reference should be added.

4- As stated by authors, IR is the strongest pathophysiological link between NAFL/MAFLD and Metabolic Syndrome. Recent studies have shown that the reduction of IR through the pharmacological eradication of HCV by direct-acting antivirals leads to both a reduction in the onset of type 2 diabetes (Diabetes, Obesity and Metabolism, 2020, 22(12):2408–2416. doi: 10.1111/dom.14168) and clinical expressions of atherosclerosis (Atherosclerosis, 2020, 296:40–47. doi: 10.1016/j.atherosclerosis.2020.01.010 - Nutrition, Metabolism & Cardiovascular Diseases (2021) 31, 2345e2353. doi: 10.1016/j.numecd.2021.04.016). These interesting issues as well as the above references deserve to be commented in discussion.

5- NAFLD/MAFLD and IR are bidirectionally correlate. Two very recent reviews explain in an updated and complete way the pathophysiological mechanisms that support this relationship (Antioxidants, 2021, 10(2), pp. 1–25, 270. doi: 10.3390/antiox10020270. - Processes, 2021, 9(1), pp. 1–18, 135. doi: 10.3390/pr9010135). Above updated references should be added to the manuscript.

6- In conclusion, the authors state “Our data could also underline how precious is to prevent NAFLD with more educational interventions to explain the importance of observing a healthy dietary regimen.” This statement, while correct, is not a direct result of this study which recruited women on average with consistently high BMI. The prevention of NAFLD should begin already in subjects with overweight, or with an initial metabolic syndrome. This message should be expressed more clearly by the authors in the conclusion.

Author Response

FIRST REVIEWER

 Thank you for your revision

 Comments and replies:

  • The population studied has an average BMI of 39.17, a borderline value between second- and third-degree obesity. Therefore, the results described by the authors on the favorable effect of a lifestyle modification on NAFLD, although very interesting, are not generalizable to a population with a lower BMI.

In accordance with the suggestions of the Reviewer, we have added in Discussion: "Our subjects in fact have an average BMI of 39.17, a borderline value between second- and third-degree obesity”.

  • Neither the text nor the tables indicate how many subjects recruited at the baseline were diabetic. Please, add this information, and, if possible, HbA1c values both at the baseline and after the lifestyle intervention. Moreover, add a comment if appropriate.

“None of the subjects recruited at the baseline were diabetic”. This sentence was added in Methods.

  • About that, it was recently observed by liver biopsy that steatohepatitis represents the sole feature of liver damage in type 2 diabetes (PLoS One. 2017 Jun 1;12(6):e0178473. doi: 10.1371/journal.pone.0178473.). This observation confirms the hypothesis that T2D and IR status increase the risk of advanced fibrosis, with the consequent worsening of hepatic outcomes. This important issue should be commented on in the discussion, and the above reference should be added.

We have added this useful information in Discussion and updated the references.

  • As stated by authors, IR is the strongest pathophysiological link between NAFL/MAFLD and Metabolic Syndrome. Recent studies have shown that the reduction of IR through the pharmacological eradication of HCV by direct-acting antivirals leads to both a reduction in the onset of type 2 diabetes (Diabetes, Obesity and Metabolism, 2020, 22(12):2408–2416. doi: 10.1111/dom.14168) and clinical expressions of atherosclerosis (Atherosclerosis, 2020, 296:40–47. doi: 10.1016/j.atherosclerosis.2020.01.010 - Nutrition, Metabolism & Cardiovascular Diseases (2021) 31, 2345e2353. doi: 10.1016/j.numecd.2021.04.016). These interesting issues as well as the above references deserve to be commented in discussion.

We have added this useful information in Discussion and updated the references.

  • NAFLD/MAFLD and IR are bidirectionally correlate. Two very recent reviews explain in an updated and complete way the pathophysiological mechanisms that support this relationship (Antioxidants, 2021, 10(2), pp. 1–25, 270. doi: 10.3390/antiox10020270. - Processes, 2021, 9(1), pp. 1–18, 135. doi: 10.3390/pr9010135). Above updated references should be added to the manuscript.

We have added this useful information in Discussion and updated the references.

  • In conclusion, the authors state “Our data could also underline how precious is to prevent NAFLD with more educational interventions to explain the importance of observing a healthy dietary regimen.” This statement, while correct, is not a direct result of this study which recruited women on average with consistently high BMI. The prevention of NAFLD should begin already in subjects with overweight, or with an initial metabolic syndrome. This message should be expressed more clearly by the authors in the conclusion.

We have added this useful information in Conclusion and updated the references.

Reviewer 2 Report

This paper aimed to evaluate the effects of lifestyle and nutritional interventions and body weight loss on hepatic steatosis through ultrasonographic and elastosonographic techniques.

METHODS and Results

  1. We know that the inclusion criteria include women with metabolic syndrome. Are all the women need to have positive NAFLD diagnosis (from Table 2 steatosis grade)?
  2. How can we know that all women keep Lifestyle modifications thought entire study (3 months)? What is the following strategy to make sure?
  3. What are the drugs that related to hyperlipidemia, hyperglycemia, and body weight change that confound the lifestyle intervention in this study?
  4. Is pair-T test considered other than dependent t-test in this study (repeated measure)?
  5. What is the percent of NAFLD grade before and after intervention in this study? How many percent of women increase or decrease NAFLD grade after the lifestyle intervention?
  6. What is the percent of fibrosis stage change in this study (before and after intervention)? We known kpa more than 14 is corresponding to liver cirrhosis in NAFLD patients.
  7. What is the percent of metabolic syndrome change after intervention?

DISCUSSION

  1. Why the factors become not significant after intervention in Linear regressions (Hip Circumference, age, Waist Circumference, Neck Circumference, WhtR, Fat Mass, GammaGT, ALT, FBG)? Mainly because of intervention? Why some factors not change of significant after intervention?
  2. Are any women not improved of NAFLD or metabolic syndrome after intervention? What is the possible reason?

Author Response

 SECOND REVIEWER

Thank you for your revision

 Comments and replies:

  • We know that the inclusion criteria include women with metabolic syndrome. Are all the women need to have positive NAFLD diagnosis (from Table 2 steatosis grade)?

“The diagnosis of NAFLD was not a criterion for inclusion”. This sentence was added in Methods.

  • How can we know that all women keep Lifestyle modifications thought entire study (3 months)? What is the following strategy to make sure?

“All subjects reported having observed lifestyle modifications”. This sentence was added in Results.

  • What are the drugs that related to hyperlipidemia, hyperglycemia, and body weight change that confound the lifestyle intervention in this study?

“None of the subjects took antidiabetic and / or lipid-lowering drugs, nor a low-calorie diet”. This sentence was added in Methods.

  • Is pair-T test considered other than dependent t-test in this study (repeated measure)?

The paired-t-test is similar to the dependent-t-test. We have corrected the text, following the suggestions of the Reviewer.

  • What is the percent of NAFLD grade before and after intervention in this study? How many percent of women increase or decrease NAFLD grade after the lifestyle intervention?

“The NAFLD was present in 88.46% before and 80.76% after intervention”.  This sentence was added in the text.

  • What is the percent of fibrosis stage change in this study (before and after intervention)? We known kpa more than 14 is corresponding to liver cirrhosis in NAFLD patients.

“None of the subjects had cirrhosis before and after 3 months. In 4 women (15.38%) the degree of fibrosis remained unchanged and in 22 subjects (84.61%) there was a decrease in the degree of fibrosis after the intervention”. This sentence was added in Results.

  • What is the percent of metabolic syndrome change after intervention?

“After 3 months, 12 subjects (46.15%) no longer presented Metabolic Syndrome. In particular, after 3 months of lifestyle modifications: 4 obese women became overweight, 17 women had BP levels <140/90 mmHg, 12 subjects showed normal glycemic levels and 23 subjects (88.46%) had total cholesterol levels <200 mg/dL”. These sentences were added in Results.

  • Why the factors become not significant after intervention in Linear regressions (Hip Circumference, age, Waist Circumference, Neck Circumference, WhtR, Fat Mass, GammaGT, ALT, FBG)? Mainly because of intervention? Why some factors not change of significant after intervention?

“It is certainly necessary a time of> 3 months to observe more significant results”. This sentence was added in Discussion.

  • Are any women not improved of NAFLD or metabolic syndrome after intervention? What is the possible reason?

“After 3 months 21 subjects still had NAFLD and 14 women still had Metabolic Syndrome”.

 This sentence was added in Results.

Round 2

Reviewer 1 Report

No further comments.

Author Response

Thank you for your revision.